# Prevalence and Numbers of Diabetes Patients with Elevated BMI in China: Evidence from a Nationally Representative Cross-Sectional Study

**DOI:** 10.3390/ijerph19052989

**Published:** 2022-03-04

**Authors:** Yongjuan Wang, Xuanyi Liang, Ziai Zhou, Zeyi Hou, Jinyu Yang, Yanpei Gao, Chenyu Yang, Tao Chen, Chao Li

**Affiliations:** 1Department of Epidemiology and Biostatistics, School of Public Health, Xi’an Jiaotong University Health Science Center, Xi’an 710061, China; wangyongjuan@stu.xjtu.edu.cn (Y.W.); xuanyi9711@stu.xjtu.edu.cn (X.L.); zhouziai@stu.xjtu.edu.cn (Z.Z.); 3120315056@stu.xjtu.edu.cn (Z.H.); yangjinyu0816@stu.xjtu.edu.cn (J.Y.); gaoyanpei1999@stu.xjtu.edu.cn (Y.G.); yangchenyu@stu.xjtu.edu.cn (C.Y.); 2Department of Public Health, Policy & Systems, Institute of Population Health, Whelan Building, Quadrangle, The University of Liverpool, Liverpool L69 3GB, UK; tao.chen@liverpool.ac.uk

**Keywords:** diabetes combined with elevated BMI, prevalence, antidiabetic medication

## Abstract

**Background:** China is facing the challenges of the increasing burden of diabetes and obesity; the prevalence and numbers of diabetes patients with obesity or overweight are still unclear. **Methods:** Nationally representative data from the China Health and Retirement Longitudinal Study (CHARLS) were used to estimate the prevalence of diabetes patients with elevated BMI, the recommendation rate for antidiabetic medication, the blood glucose control rate, and the corresponding population size. **Results:** The prevalence of diabetes patients with elevated BMI was 9.18% (95% CI: 7.88, 10.68; representing 31.54 million) in China. More than half of people with diabetes had elevated BMI (overweight or obesity). Among the participants who were not taking antidiabetic medication, 26.15% (95% CI: 18.00, 36.36; representing 3.79 million) were recommended for antidiabetic medication by the 2020 CDS guideline. There were 24.62% (95% CI: 16.88, 34.45; representing 3.64 million) patients, representing 11.13 (95% CI: 9.86, 12.41) million people, with diabetes combined with elevated BMI, taking antidiabetic medication, and still above the goal blood glucose. **Conclusions:** Our results indicate that diabetes combined with elevated BMI has become a major public health problem in China in people over 45 years of age. Moreover, the prevalence and population size of women are higher than those of men, and the prevalence of people over 65 years old is slightly lower than that of elderly people aged 45–65. The recommended rate of antidiabetic medication and the control rate of blood glucose were high, and prevention and treatment strategies for diabetes combined with elevated BMI are needed.

## 1. Introduction

Diabetes mellitus (DM) is a group of metabolic diseases characterized by chronic hyperglycemia caused by disorders of insulin secretion and/or utilization caused by multiple etiologies [1]. Type 2 diabetes mellitus can cause a series of complications, mainly increasing the heart disease risk of vascular disease and diabetic eye disease, diabetic foot, etc. [2]. Currently, several common non-communicable diseases are on the rise globally, and diabetes is no exception [3]. In 1995, the global adult diabetes prevalence rate was 4.0% [4]. A study from the *International Diabetes Federation Diabetes Atlas 9th edition* [5] showed that the global prevalence of diabetes in 2019 was estimated to be 9.3%, representing 463 million people with diabetes worldwide, and it is predicted that the prevalence will rise to 10.2% (578 million) and 10.9% (700 million) in 2030 and 2045, respectively. Half (50.1%) of people do not even know if they are diabetic, which greatly increases the burden of global disease.

At present, diabetes and abnormal weight are the main public health problems in China. A study published in *JAMA* shows that the prevalence of diabetes in China has risen rapidly from 0.67% in 1980 to 10.9% in 2013 [6]. Additionally, the numbers of overweight and obese in China continued to increase and within two decades, from 1993 to 2015, and the burden reached the levels of 41% for overweight, 15% for obesity, and 47% for abdominal obesity, based on the China Health and Nutrition Survey (CHNS) [7]. The relationship between diabetes and obesity has also been recognized. A study published by an international working group composed of 32 medical experts showed that most diabetic patients were overweight or obese and explained that the sharp increase in the prevalence of diabetes in the past 20 years is precisely because of the obesity epidemic [8]. The reason is that almost 90% of type 2 diabetes can be attributed to being overweight [9]. Compared with adults of normal weight, people with a BMI > 40 are diagnosed with diabetes at an OR value of 7.37 (95% confidence interval (CI): 6.39–8.5) [10].

Some studies have estimated the prevalence of diabetes and obesity in China; however, there are still some limitations [6,11,12,13,14,15]. First, several estimates of the prevalence and population size of diabetes in China are not based on China’s diabetes guidelines and diagnostic criteria. The two surveys in 2010 [11] and 2013 [6] adopted the 2010 standards defined by the American Diabetes Association (ADA), while the surveys by Pan et al. [12] and Yang et al. [13] adopted the more stringent standards of the World Health Organization (WHO). Therefore, the situation in China is estimated to be biased. Second, because some studies did not use nationally representative data, the results can only represent certain regions [14,15]. Most importantly, no studies have estimated the prevalence and population size of people with diabetes combined with abnormal body weight in China.

Our study aimed to estimate (1) the prevalence and the population size of diabetes with elevated BMI patients, (2) candidacy for initiation of antidiabetic treatment, and (3) who with HBA1c or blood glucose above their goals are among the diabetes patients with elevated BMI and taking antidiabetic medication by using the nationally representative data. 

## 2. Methods

### 2.1. Data Sources and Study Population

The China Health and Retirement Longitudinal Study (CHARLS) is a survey of middle-aged and elderly people in China over 45 years old. We used the 2011–2012 CHARLS national baseline survey data to estimate the population-level impact on prevalence or numbers of diabetes combined with elevated BMI and recommendation for antidiabetic medication according to the Guidelines for the prevention and treatment of type 2 diabetes mellitus in China (2020 edition). The profile and data quality of CHARLS had been reported previously [16]. In brief, CHARLS (2011–2012) uses a multi-stage sample of 150 randomly sampled county-level units, excluding Tibet. In addition, there were some missing values in the blood examination indicator. However, when calculating the sample weights, we used the personal weights provided by the CHARLS database for physical examination data to correct for the physical examination non-response. All sample weights are directly constructed by sampling probability to ensure the accuracy of the estimation. The data of this study can be obtained from http://charls.pku.edu.cn/index/en.html (accessed on 8 October 2021).

Written informed consent was obtained from all participants of the CHARLS, and this study was approved by the institutional review board. This study was also approved by the Human Research Ethics Committee of the Xi’an Jiaotong University Health Science Center (No: 2021-6).

### 2.2. Blood Glucose Measurement and Definition

The blood samples were collected from three veins from each participant by medically trained staff from the Chinese Center for Disease Control and Prevention (CDC), and participants were asked by the professional staff to fast at night in accordance with the standard protocol. The glycosylated hemoglobin (HbA1c) and glucose were measured from frozen plasma or whole blood at the laboratory [16]. In this study, we focused on type 2 diabetes, excluding type 1 diabetes and secondary diabetes. Diabetes was defined as a self-reported history of diabetes, current or former use of diabetic medications, fasting blood glucose ≥ 7.0 mmol/L, random blood glucose ≥ 11.1 mmol/L, or glycosylated hemoglobin ≥ 6.5% (HbA1c ≥ 6.5%). The 2020 CDS guidelines recommended initiating antidiabetic medication for diabetic patients with HbA1c ≥ 7% and a treatment goal of HbA1c < 7% [17].

### 2.3. Elevated BMI

Height and weight indicators were measured as CHARLS non-blood biomarkers. Participants took off their shoes and stood on the treads of the Seca^TM^213 altimeter manufactured by Seca (Hangzhou) Co., Ltd. with their backs to a post. A staff member slid down the headboard and gently touched the interviewee’s head to record readings. To measure their weight on the Omron^TM^ HN-286 weighing scale by Ke Rui Er Science and Technology (Yang Zhou) Co., Ltd., participants removed their shoes, heavy coats, and heavy objects in their pockets and stood on the scale until their weight was displayed on a screen and recorded by a worker. The body mass index (BMI) is the metric currently in use for defining anthropometric height/weight characteristics in adults and for classifying (categorizing) them into groups. Body mass index (BMI) was calculated as weight in kilograms divided by height in meters squared (kg/m^2^). Overweight in this study was defined as 28 kg/m^2^ > BMI ≥ 24 kg/m^2^, and obesity was defined as BMI ≥ 28 kg/m^2^ [18]. Elevated BMI includes overweight and obesity.

### 2.4. Statistical Analysis

The percentage and number (95% confidence interval (CI)) of adults with diabetes, elevated BMI, diabetes combined with elevated BMI, and recommended antidiabetic treatment and control status of diabetes combined with elevated BMI, based on the 2020 CDS guideline, were estimated using CHARLS (2011–2012) sampling weights to extrapolate to the Chinese population (≥45 years). 

The baseline characteristics, including the categorical variables, referred to in this study included age (45–59, 60–74, ≥75), gender (male or female), educational level (illiterate, primary school, middle/high school, bachelor, or above), marital status (married or never married), registered residence (rural or urban), smoking status (non-smoker, current smoker, or ex-smoker), and drinking status (non-drinker or drinker). Continuous variables were expressed as mean ± standard deviation (SD) and classified variables were expressed as N (%) between each group. Baseline characteristics of participants from CHARLS (2011–2012) were compared between diabetes, elevated BMI, and diabetes combined with elevated BMI groups by variance analysis for continuous variables expressed as mean (standard deviation) or Chi-square tests for categorical variables expressed as *n* (%). All reported *p*-values are two-sided, and statistical analyses were conducted with Stata 15.0 statistical software (StataCorp LLC, College Station, TX, USA).

## 3. Results

### 3.1. Prevalence and Numbers of Diabetes Patients with Elevated BMI, Recommended to Initiate Antidiabetic Medication, or above Goal Blood Glucose

Overall, we estimate that the prevalence (95% CI) of diabetes in adults over 45 years of age in China is 6.64% (5.90, 7.47), representing 22.8 (95% CI: 20.57, 25.04) million people. There are 31.54 million people with diabetes combined with elevated BMI, accounting for 9.18% (95% CI: 7.88, 10.68), indicating that more than half of all diabetic patients (with or without elevated BMI) are overweight or obese. Table 1 shows that the prevalence and population size of females with diabetes combined with elevated BMI is higher than in males, representing 18.61 million (in females) and 12.93 million (in males). The prevalence of diabetes combined with elevated BMI is higher in people under 65 years old (9.44%, representing 23.57 million). The prevalence of isolated diabetes is higher in males (7.04%, representing 11.50 million) and people over 65 years of age (9.44%, representing 8.85 million).

Among participants who reported not taking antidiabetic medication, 26.15% (95% CI: 18.00, 36.36; representing 3.79 million) were recommended antidiabetic medication by the 2020 CDS guideline; however, 73.85% (95% CI: 63.64, 82.00; representing 12.98 million) were not recommended initiating antidiabetic medication, as in Figure 1b and Figure 2. Among those who had diabetes combined with elevated BMI and taking antidiabetic medication, 75.38% (95% CI: 65.55, 83.12) had below goal blood glucose, representing 11.13 (95% CI: 9.86, 12.41) million people in China, and 24.62% (95% CI: 16.88, 34.45; representing 3.64 million) were still above goal blood glucose after taking antidiabetic medication. (Figure 1c and Figure 2) Stratified by age and sex, the recommended antidiabetic medication rate was higher for aged ≥ 65 and female, and the control rate of blood glucose was better (Appendix A Table A1).

### 3.2. Baseline Information of Participants Included in CHARLS 2011–2012 Study

A total of 9410 participants were included in the present analysis and we divided the participants into four categories: normal blood glucose and BMI (*n* = 4904), isolated elevated BMI (*n* = 3070), isolated diabetes (*n* = 658), and diabetes combined with elevated BMI (*n* = 778). The average age of people who had diabetes combined with elevated BMI was 59.17 ± 8.48 (mean ± SD) and 58.10% were male. The BMI for the diabetes combined with elevated BMI group was 27.71 kg/m^2^ (SD = 3.85), with SBP 138.16 mm Hg (SD = 21.00) and DBP 80.03 mm Hg (SD = 11.51), respectively. In addition, these subjects were characterized as follows according to different stratification conditions: The percentage of females is higher (58.10%) than males (41.90%). For educational level, the proportion of primary school (38.48%) is slightly higher than that of middle/high school (33.42%). Registered residences are mainly rural, accounting for 73.78%. The proportions of non-drinking and non-smoking patients were higher, accounting for 71.85% and 77.51%, respectively (Table 2). Statistical descriptions of descriptive and categorical variables are provided in Table A2 .

## 4. Discussion

Our study estimated the prevalence and population size of diabetes combined with elevated BMI in China and described the characteristics of this population. Specifically, there are 31.54 million patients who have both diabetes and high BMI, and the prevalence rate is 9.18%. To the best of our knowledge, no study has reported the prevalence and population size of diabetes combined with elevated BMI. We found that diabetes patients with high BMI accounted for more than half of the total diabetic patients (31.54 million/54.34 million) and even exceeded the number of people with diabetes alone. This means that 57.67% of all diabetic patients have high BMI (overweight or obese).

A 2013 study in China showed that the prevalence of overweight/obesity accounted for 41.0% and 24.3% [19]. Overweight and obesity are risk factors for diabetes that might be explained by the insulin resistance, type 2 diabetes in obese people, insulin sensitivity, and β-islet cells’ adjust ability declining. A decrease in β-islet cells’ function is one of the leading causes of type 2 diabetes, which means almost 90% of type 2 diabetes can be attributed to being overweight or obese [20,21]. In addition, our study found that diabetes combined with elevated BMI had significantly higher blood pressure than the other three groups. Obesity is also an important risk factor for hypertension and high cholesterol [22], the “three high diseases” very common in China, which means that the prevalence of type 2 diabetes will continue to rise with the increase in overweight and obesity.

In 2019, the International Diabetes Federation (IDF) released the 9th edition of the Global Diabetes Map, the data estimated that nearly 19.9% of elderly people with diabetes (65 years of age and older) worldwide and the prevalence of diabetes increases with age [5]. According to the latest diabetes diagnostic criteria in 2020, our research also found that the prevalence rate of diabetes in adults over 65 in China is 17.94%, representing 16.79 million people. From the perspective of gender stratification, our findings are consistent with those of Pradhan, A.D et al. [23]. Our study also has a higher prevalence of women (10.33%, representing 18.61 million) with diabetes and high BMI than men (7.92%, representing 12.93 million). A study by Hu L et al. [24] showed that, with an increase in age, the prevalence of overweight and obesity increased in both men and women. The relative risk of diabetes with abnormal weight varies depending on certain characteristics. The relative risk of diabetes in obese women is eight times higher than in normal-weight women, while the risk of diabetes in obese men is six times higher than in normal-weight men. It can be seen that both overweight and obesity have a significant risk of diabetes, that is, people with abnormal weight have a higher risk of diabetes than the general population. This is consistent with our findings. 

Our results also showed that 26.15% of patients with diabetes combined with elevated BMI were recommended to take antidiabetic medication. Among patients with diabetes and elevated BMI who are actively taking antidiabetic medication, there are still 24.62% of patients with blood glucose higher than the target value. As we all know, obesity can aggravate the insulin resistance of diabetic patients, thus increasing the difficulty of blood glucose control, and the insulin resistance of diabetic patients also further increases the difficulty of weight loss, resulting in a vicious circle that leads to many patients who receive antidiabetic medication treatment and have not reached the blood sugar control standard. This is consistent with the results of other studies [11,21]. The 2020 CDS guideline [17] emphasizes the weight management goals, drug selection, and metabolic surgery of patients with diabetes associated with overweight/obesity. Although the 2020 CDS guideline suggests that these people should actively carry out lifestyle intervention and drug therapy and carry out weight loss surgery when necessary, the prevalence of diabetes mellitus combined with BMI is still high. The 2013 China Chronic Disease Risk Factor Surveillance Report [25] showed that the awareness rate of diabetes among adults in China was 38.6% and was higher in urban areas (45.3%) than in rural areas (31.1%). This is similar to the results of our study. Table 2 shows that the number of patients in rural areas is 2.8 times that of urban patients and rural patients generally have a low level of education. Diabetes and obesity are chronic diseases, patients are not sufficiently vigilant and attentive, and the lifestyle interventions carried out by patients and their families are not compliant and lack executive power [26]. Patients with diabetes combined with elevated BMI not only experience difficulties in weight management and blood glucose control but also increased risk of hypertension, dyslipidemia, and cardiovascular and cerebrovascular events. Our results indicate that China is still facing severe challenges.

This study has several limitations that need to be explained. First, the data in this study only included the data of people over 45 years old, so it was impossible to estimate the prevalence rate and population size of people under 45 years old, and the age results could not be extrapolated to the whole population. Second, when baseline data were collected in the database used in this study, there were missing values of HbA1c and blood glucose, but the proportion of missing values was not large. The CHARLS database also gave a weight estimation of missing values of blood, so our research results were reliable. Third, the use of data from 2011 to 2012 in this study underestimates the current prevalence and population size of diabetes with elevated BMI over the age of 45 in China using the 2020 CDS standards. The actual situation may be even worse.

## 5. Conclusions

Overall, our findings indicate that diabetes with elevated BMI is common in China. Moreover, there are still a number of diabetes patients with elevated BMI who are recommended to take antidiabetic medication and fail to achieve blood glucose control standards. These results indicate that elevated BMI diabetes has become a public health challenge in China and highlight the need for national strategies to prevent, detect, and treat elevated BMI diabetes in people over 45 years of age.

## Figures and Tables

**Figure 1 ijerph-19-02989-f001:**
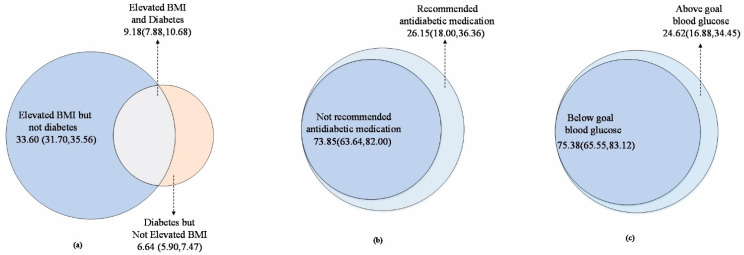
Data used for estimating were from CHARLS 2011–2012 baseline survey. (**a**) Percentage of Chinese adults who have elevated BMI, diabetes, or both. (**b**) Proportion of patients who are not taking antidiabetic medication and who have diabetes combined with elevated BMI. (**c**) Proportion of people taking antidiabetic medication with diabetes combined with elevated BMI and whether their blood glucose reaches the target.

**Figure 2 ijerph-19-02989-f002:**
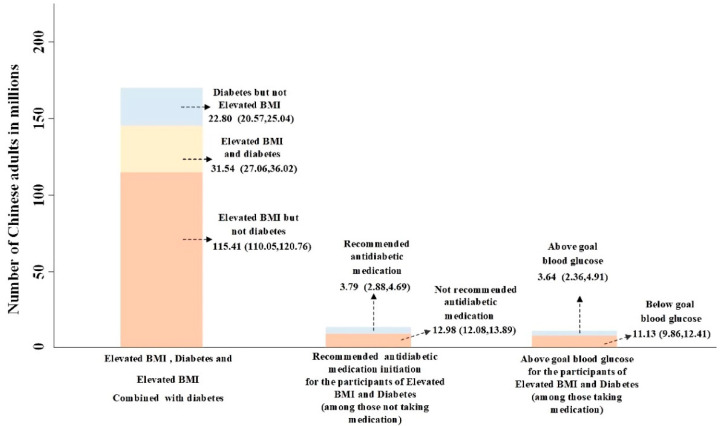
Number of Chinese adults who have elevated BMI, diabetes, or both, who are recommended initiating antidiabetic medication among those not taking antidiabetic medication and those with diabetes combined with elevated BMI or with above goal blood glucose among those taking antidiabetic medication. Data used for estimating were from the CHARLS 2011–2012 baseline survey.

**Table 1 ijerph-19-02989-t001:** Percentage and numbers of Chinese adults who have elevated BMI ^2^, diabetes ^3^, or both.

Description	Normal ^1^		Isolated Elevated BMI		Isolated Diabetes		Elevated BMI and Diabetes	
	Percentage	Numbers (95% CI)	Percentage	Numbers (95% CI)	Percentage	Numbers (95% CI)	Percentage	Numbers (95% CI)
Total	50.57	173.68	33.60	115.41	6.64	22.80	9.18	31.54
	(48.30,52.84)	(168.07,179.30)	(31.70,35.56)	(110.05,120.76)	(5.90,7.47)	(20.57,25.04)	(7.88,10.68)	(27.06,36.02)
Age < 65	48.66	121.50	36.31	90.65	5.59	13.96	9.44	23.57
	(46.23,51.10)	(116.62,126.38)	(34.52,38.13)	(86.02,95.28)	(4.91,6.36)	(12.34,15.58)	(7.86,11.31)	(19.43,27.71)
Age ≥ 65	55.66	52.18	26.41	24.76	9.44	8.85	8.50	7.97
	(52.01,59.24)	(49.38,54.98)	(22.84,30.32)	21.97,27.54)	(7.86,11.30)	(7.33,10.36)	(6.84,10.52)	(6.27,9.67)
Male	57.11	93.26	27.93	45.62	7.04	11.50	7.92	12.93
	(54.58,59.61)	(89.53,96.98)	(25.88,30.09)	(42.32,48.92)	(6.02,8.22)	(9.81,13.19)	(6.71,9.32)	(10.97,14.88)
Female	44.65	80.42	38.74	69.79	6.28	11.30	10.33	18.61
	(41.73,47.60)	(76.56,84.29)	(36.23,41.32)	(65.70,73.88)	(5.42,7.25)	(9.85,12.76)	(8.38,12.67)	(14.62,22.61)

Baseline data from the 2011–2012 China Health and Retirement Longitudinal Study (CHARLS) were used to estimate the percentage and numbers of Chinese adults. ^1^ Normal is defined as neither diabetes nor elevated BMI. ^2^ Elevated BMI: body mass index ≥ 24.0 kg/m^2^. ^3^ Diabetes was defined as fasting blood glucose ≥ 7.1 mmol/L or any blood glucose ≥ 11.1 mmol/L, and HbA1c ≥ 6.5%.

**Table 2 ijerph-19-02989-t002:** Baseline characteristics of the study population of Chinese adults and the presence of elevated BMI ^1^, diabetes ^2^, or both.

	Normal	Isolated Elevated BMI	Isolated Diabetes	Elevated BMI and Diabetes	Total	*p* Value
Number of participants	4904	3070	658	778	9410	
Age, mean ± SD	60.47 ± 9.71	57.54 ± 8.73	62.19 ± 9.55	59.17 ± 8.48	59.53 ± 9.41	<0.001
Age						<0.001
45–59	2582 (52.65)	1943 (63.29)	300 (45.59)	439 (56.43)	5264 (55.94)	
60–74	1884 (38.42)	1002 (32.64)	279 (42.40)	303 (38.95)	3468 (36.85)	
≥75	438 (8.93)	125 (4.07)	79 (12.01)	36 (4.63)	678 (7.21)	
Gender						<0.001
Male	2576 (52.53)	1157 (37.69)	318 (48.33)	326 (41.90)	4377 (46.51)	
Female	2328 (47.47)	1913 (62.31)	340 (51.67)	452 (58.10)	5033 (53.49)	
Educational level						<0.001
Illiterate	1513 (30.85)	814 (26.51)	213 (32.37)	206 (26.48)	2746 (29.18)	
Primary school	2095 (42.72)	1198 (39.02)	291 (44.22)	299 (38.48)	3883 (41.26)	
Middle/High school	1248 (25.45)	1004 (32.70)	146 (22.19)	260 (33.42)	2658 (28.25)	
Bachelor or above	48 (0.98)	54 (1.76)	8 (1.22)	13 (1.67)	123 (1.31)	
Marital status						
Never	51 (1.04)	8 (0.26)	5 (0.76)	2 (0.26)	66 (0.70)	<0.001
Married	4853 (98.96)	3062 (99.74)	653 (99.24)	776 (99.74)	9344 (99.30)	
Registered residence						<0.001
Rural	4268 (87.05)	2405 (78.39)	548 (83.28)	574 (73.78)	7795 (82.86)	
Urban	635 (12.95)	663 (21.61)	110 (16.72)	204 (26.22)	1612 (17.14)	
Smoking status						<0.001
Non-smoker	3078 (62.77)	2424 (78.96)	438 (66.57)	603 (77.51)	6543 (69.53)	
Smoker	1826 (37.23)	646 (21.04)	220 (33.43)	175 (22.49)	2867 (30.47)	
Drinking status						<0.001
Non-drinker	3138 (64.07)	2198 (71.60)	430 (65.55)	558 (71.85)	6325 (67.27)	
Drinker	1760 (35.93)	872 (28.40)	226 (34.45)	219 (28.15)	3077 (32.73)	
FBG (fasting blood-glucose), mean ± SD	5.56 ± 0.78	5.71 ± 0.77	9.17 ± 4.30	8.99 ± 3.62	6.14 ± 2.11	<0.001
Glycated hemoglobin, mean ± SD	5.08 ± 0.39	5.14 ± 0.40	6.13 ± 1.66	6.31 ± 1.50	5.27 ± 0.82	<0.001
BMI, mean ± SD	20.97 ± 2.01	26.99 ± 3.00	21.32 ± 1.94	27.71 ± 3.85	23.52 ± 3.95	<0.001
SBP, mean ± SD	127.46 ± 21.25	133.51 ± 21.21	131.83 ± 21.38	138.16 ± 21.00	130.62 ± 21.52	<0.001
DBP, mean ± SD	73.49 ± 11.82	78.63 ± 12.14	74.42 ± 10.88	80.03 ± 11.51	75.77 ± 12.13	<0.001

BMI, Body Mass Index; SBP, systolic blood pressure; DBP, diastolic blood pressure. Elevated BMI includes overweight and obesity. ^1^ Normal was defined as having neither diabetes nor elevated BMI. ^2^ Elevated BMI: body mass index ≥ 24.0 kg/m^2^.

## Data Availability

The CHARLS data that support the findings of this study are available upon application from http://charls.pku.edu.cn/index/en.html (accessed on 8 October 2021).

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
