# Peer review of "Prevalence and Numbers of Diabetes Patients with Elevated BMI in China: Evidence from a Nationally Representative Cross-Sectional Study"

_ijerph, 2022, doi:10.3390/ijerph19052989_

Round 1
Reviewer 1 Report
Many thanks for the opportunity to review the article „Prevalence and numbers of diabetes patients with elevated BMI in China: evidence from a nationally representative cross sectional study”.
- The article has been greatly improved, but there are still editorial errors: lack of text alignment in some places (for example, lines 133-145), inadequate spacing (for example, lines 91-100)
- In Figure 1 and Figure 2 the inscriptions are unfortunately still illegible.
- In my opinion, each BMI value appearing in the text of the paper should be accompanied by a unit - kg/m2.
Reviewer 2 Report
Diabetes and obesity have become the main public health problems in the world including China.
This study aims to estimate the prevalence of diabetes patients with elevated BMI.
The authors concluded that diabetes combined with elevated BMI has become a major public health problem in China in people over 45 years of age.
This article is well written and of clinical interest.
However, several issues should be improved before the consideration for publication.
Major comments
1 It is unclear what types of diabetes the authors mentioned. Considering the criteria for diabetes (Line 97), type 1 and secondary diabetes were likely to be included, which should be clearly described. The incidents of Type 1 and secondary diabetes are usually unrelated to the obesity.
2 It may be better to indicate the prevalence of diabetes patients with elevated BMI, at least, according to men and women. In addition, I wonder the prevalence may be lower in the elderly, although the authors described “major public health problem in China in people over 45 years of age” (Line 24).
3 I would like to know severity and complications in such diabetes patients with elevated BMI (Table 2).
4 It is hard to see the contents of Figures. Larger fonts and clear figures should be provided.
5 Do you have the data of HbA1c and fasting plasma glucose? If so, please add these data in Table 2, which is informative for the readers.
6 I wonder age difference rather than BMI may influence the incident of diabetes in four groups in table 2. Therefore, age should be adjusted between groups in the statistical analysis.
Sex is also to be adjusted preferably.
7 Is the order for four groups in table 2 validated? If not, post-hoc test is preferably to be added.
Round 2
Reviewer 2 Report
Thank you for interesting paper.
I confirmed the manuscript has been improved.
I have nothing to comment.
This manuscript is a resubmission of an earlier submission. The following is a list of the peer review reports and author responses from that submission.
Round 1
Reviewer 1 Report
Introduction:
Please carefully read the entire introduction section as there are certain language issues. Also, some citations are missing. For example, sentences 29-30, has no citation. Lines 32-33, which study shows? be specific. What is the relationship between diabetes and obesity? Is obesity is a predisposing factor or an etiologic factor? Explain in the text. Literature Review is missing here. What is the status of existing literature? What is the similarity (difference) of your study with the past literature? What are the key contributions of your study? Please explicitly provide description of result in one paragraph in the introduction section.
Method:
Are you using a cross sectional dataset? If so, then be careful with your wordings. You are not estimating the impact. You just discover the correlation between variables. What is CDC stands for? I know it but readers might not know that. Please be careful with your writings.
What is the specific estimation model in this analysis? Regression (Logit, Probit or any other)?
Results:
Your results here is listed as a grocery list. Please consulate with someone who has knowledge of academic writing. The result section is full name numbers which makes is difficult to follow. Rewrite the entire result section.
Please provide a Descriptive stat table for all the variables in the analysis.
Conclusions:
Too short and many things need to be explained.
Reviewer 2 Report
The article submitted for review is first and foremost not editorially polished.
Below are my comments and suggestions:
- Review the literature record in the text of the paper - it does not conform to the requirements of the journal
- BMI values are missing the unit (kg/m2) - for example line 86, 87, 130, please add throughout the paper
- When naming equipment, for example for measuring weight and height, provide details (e. g. place of manufacture)
- Figure 1 is not very readable, please edit
- Table 1 is not very readable, please edit
- Under Tables 1 and 2, define the word "normal"
- In the text of the article, reference is first made to Table 3 and then to Table 2. The order should follow the appearance of tables in the text of the paper.
- Different font sizes appear in the text of the paper - this should be standardized according to the requirements of the journal
- Figure 1 is not very readable, please edit
- Line 154-156 - in my opinion it is not necessary to give the following notation, for example: 4904/9410, 4904 people will suffice. Similarly in the other groups.
- Description of results presented too chaotically.
- The discussion does not cover all aspects raised in the discussion of the results. Among other things, there is a lack of reference to gender.